# Impact of Isoniazid Preventive Therapy on Tuberculosis incidence among people living with HIV: A secondary data analysis using Inverse Probability Weighting of individuals attending HIV care and treatment clinics in Tanzania

Werner M. Maokola[1,2]*, Bernard J. Ngowi[3], Michael J. Mahande[2], Jim Todd[4,5], Masanja Robert[6], Sia E. Msuya[2]

1 Ministry of Health, Community Development, Gender, Elderly and Children, Dar es Salaam, Tanzania, 2 Institute of Public Health, Kilimanjaro Christian Medical University College, Moshi, Tanzania, 3 Mbeya University College of Health Sciences, Mbeya, Tanzania, 4 National Institute of Medical Research, Mwanza, Tanzania, 5 London School of Hygiene and Tropical Medicine, London, United Kingdom, 6 Mwenge Catholic University, Moshi, Tanzania

* drwernerm@yahoo.com

## Abstract

### Background

Information on how well Isoniazid Preventive Therapy (IPT) works on reducing TB incidence among people living with HIV (PLHIV) in routine settings using robust statistical methods to establish causality in observational studies is scarce.

### Objectives

To evaluate the effectiveness of IPT in routine clinical settings by comparing TB incidence between IPT and non-IPT groups.

### Methods

We used data from PLHIV enrolled in 315 HIV care and treatment clinic from January 2012 to December 2016. We used Inverse Probability of Treatment Weighting to adjust for the probability of receiving IPT; balancing the baseline covariates between IPT and non-IPT groups. The effectiveness of IPT on TB incidence was estimated using Cox regression using the weighted sample.

### Results

Of 171,743 PLHIV enrolled in the clinics over the five years, 10,326 (6.01%) were excluded leaving 161,417 available for the analysis. Of the 24,800 who received IPT, 1.00% developed TB disease whereas of the 136,617 who never received IPT 6,085 (4.98%) developed

**Data Availability Statement:** All relevant data are within the paper and its S1 Dataset.

**Funding:** The study was funded through SEARCH (Sustainable Evaluation through Analysis of Routinely Collected HIV data) Project under Bill and Melinda Gates Foundation grant number OPP1084472 entitled "Using routinely collected public facility data for program improvement in Tanzania, Malawi and Zambia.". Routine HIV data management is Tanzania is managed collaboratively by the Government of the Republic of Tanzania and the Government of the United States of America through President's Emergency Plan for AIDS Relief.

**Competing interests:** This study was funded through the SEARCH Project under Bill and Melinda Gates Foundation. Routine HIV data is managed by the Government of the Republic of Tanzania in collaboration with the Government of the United STates of America through President's EMergency Plan for AIDS Relief (PEPFAR).

**Abbreviations:** ART, Antiretroviral Treatment; ARV, Antiretroviral Therapy; ATE, Average Treatment Effect; BMI, Body Mass Index; CI, Confidence Interval; CTC, Care and Treatment Clinic; HR, Hazard Ratio; ICF, Intensified TB Case Finding; IPT, Isoniazid Preventive Therapy; IPTW, Inverse Probability of Treatment Weighting; NIMR, National Institute of Medical Research; PLHIV, People Living with HIV; TB, Tuberculosis; WHO, World Health Organization.

TB disease. In 278,545.90 person-years of follow up, a total 7,052 new TB cases were diagnosed. Using the weighted sample, the overall TB incidence was 11.57 (95% CI: 11.09–12.07) per 1,000 person-years. The TB incidence among PLHIV who received IPT was 10.49 (95% CI: 9.11–12.15) per 1,000 person-years and 12.00 (95% CI: 11.69–12.33) per 1,000 person-years in those who never received IPT. After adjusting for other covariates there was 52% lower risk of developing TB disease among those who received IPT compared to those who never received IPT: aHR = 0.48 (95% CI: 0.40–0.58, P<0.001).

## Conclusion

IPT reduced TB incidence by 52% in PLHIV attending routine CTC in Tanzania. IPTW adjusted the groups for imbalances in the covariates associated with receiving IPT to achieve comparable groups of IPT and non-IPT. This study has added evidence on the effectiveness of IPT in routine clinical settings and on the use of IPTW to determine impact of interventions in observational studies.

## Background

Tuberculosis (TB) is a common opportunistic infection among people living with HIV (PLHIV) [1, 2]. Despite strategies for control and prevention of TB among PLHIV, co-infection of TB and HIV (TBHIV) is still a public health concern [3–5]. Worldwide in 2018, 862,000 (2.3%) of all PLHIV had TB disease and 30% of all HIV deaths occurred in individuals with TBHIV co-infection [6, 7]. The African continent accounted for 24% of all TB cases diagnosed and was second to Asia in the number of TB cases. Tanzania was ranked among 30 countries in the World with high TB burden [7]. HIV is a known risk for TB especially in the absence of antiretroviral therapy (ART). The burden of TB among PLHIV in Tanzania was reported to 16.7 cases per 1000 person-years [8].

Tanzania adopted World Health Organization (WHO) three I's strategy which comprises of intensified TB Case Finding (ICF), Isoniazid Preventive Therapy (IPT) and Infection control and prevention in all clinical settings since 2010 [9]. IPT entails giving an anti-TB drug to eligible PLHIV for at least 6 months to treat latent TB infection (LTBI); individuals infected with TB but without TB disease [10, 11]. IPT integration into HIV care and treatment services in Tanzania started in 2011 starting with high level health facilities as a phased implementation to lower level health facilities in 2012, so that 50% of care and treatment clinics (CTC) was implementing IPT by the end of December 2018 [12]. However, routine data from PLHIV attending CTC in 3 regions in Tanzania documented 14% had initiated IPT from 2012–2016 [13] whereas the target was to cover a minimum of 50% of clinic attendees [14].

IPT is a proven public health intervention to reduce TB among PLHIV by treating LTBI, thus preventing LTBI from developing into TB disease. A systematic review and meta-analysis from clinical trials have demonstrated a reduction in TB incidence among PLHIV following IPT ranging from 30% to 74% [4, 15–17].

Moreover, in routine clinical settings have shown a reduction in TB incidence of between 48% and 76% in PLHIV receiving IPT [18–21]. These studies conducted in Ethiopia, Tanzania and Brazil but did not address the challenges of using observational studies to establish cause-effect relationship between IPT and TB incidence when IPT is rolled out in routine clinical settings, the baseline characteristics of those initiated on IPT may differ from those who are not

initiated on IPT. Propensity scores are one way to obtain an unbiased estimate of the effectiveness of IPT in preventing TB incidence using observational data from routine clinical settings [22]. This paper reports a secondary analysis of a cohort of PLHIV enrolled in CTC from January 2012 to December 2016 in the three regions in Tanzania. We applied Inverse-Probability for Treatment Weighting (IPTW)-one of the propensity score approaches to the data in order to balance baseline characteristics of PLHIV who received and those who never received IPT [23]. This approach obtained an objective estimate of the impact of IPT intervention on TB incidence in using observational data from routine clinical settings where randomization of the intervention is not done.

## Methods

### Study design and study population

This analysis used routine data from PLHIV enrolled in 315 CTC in three regions of Tanzania from January 2012 to December 2016.

### Study setting

More information regarding the HIV care and treatment program in Tanzania, including the integration of the TB services in HIV clinics, has been previously Published [24]. Tanzania has 26 regions and the three regions chosen were among those with the highest HIV prevalence [25]. Since 2012, all health facilities with CTC in Tanzania have implemented ICF and TB infection control and prevention, with IPT incrementally rolled out across Tanzania. During every clinic visit, PLHIV were screened for the presence of TB symptoms and signs as part of ICF. In health facilities where IPT was available, PLHIV who screened TB negative and fulfill other clinical criteria were initiated on IPT for 6 months to treat latent TB. Those who screened positive for TB underwent further TB disease diagnosis using diagnostic tests according the TB diagnosis algorithm [26]. TB infection control and prevention was another intervention to reduced TB among PLHIV and was implemented in all health facilities in the country.

### Data collection

Detailed information regarding the demographic and clinical data collected in the Tanzania CTC has been described elsewhere [13, 24, 27]. At every visit to the CTC data were entered into an electronic database which was collated on a central server at national level. Independent variables used in the study namely sex, age, WHO clinical stage, Antiretroviral Therapy (ART) status, nutritional status, weight, height, enrolment year, health facility type and ownership and region where the client was registered were routinely recorded in the database. The database also recorded information on IPT initiation and completion dates and on TB screening and management. This study extracted those demographic and clinical information including those of TB services using the unique patient identification number for linking clinic visits. Any PLHIV who had any evidence of LTBI on the first visit to CTC were excluded from the analysis.

All PLHIV attending CTCs which never implemented IPT throughout the study period was also excluded from the analysis.

### Data analysis

The primary exposure was IPT (IPT and non-IPT) use and TB disease incidence was the outcome of interest. "IPT" were individuals who were ever initiated on IPT and outcome of

interest "non-IPT" were those who were never initiated on IPT throughout the study period. Sex, age, WHO clinical stage, ART status, Body Mass Index (BMI), functional status, health facility type, health facility ownership, enrolment year and region and were the covariates needed to be adjusted for in the analysis. Characteristics of PLHIV who received IPT and those who never received IPT during the study duration were expressed using unweighted sample. To balance for measured covariates between those who received IPT and those who never received IPT, IPTW was applied. The following steps were followed in creating weighted dataset: (1) **Selection of covariates which are potential confounders (affecting both IPT initiation and TB disease diagnosis)**: Sex, age, functional status, ART status, Body Mass Index (BMI), nutritional status, health facility type, region and health facility ownership (12,19,28) were selected for propensity score model. **(2) Creating Propensity scores (probability of receiving IPT condition to covariates)**: As these covariates differed between those who received IPT and those who never received IPT. Logistic regression was used to calculate propensity score to estimate the odds of receiving IPT. From the propensity scores, IPTW was created for each PLHIV to make a synthetic sample in which IPT and non-IPT groups were balanced.

The treatment effect to be calculated is Average Treatment Effect (ATE). ATE refers to the treatment effect to the entire target population **(3) Checking for balance after weighting**: The balance between the two groups was determined quantitatively by comparing standardized means and variances of the 2 groups. Weighted sample had standardized difference of closer to zero whereas the variance in the weighted group was closer to 1 compared to unweighted sample, showing covariate balance in the weighted sample [28, 29]. **(4) Calculating Propensity Scores weights**: From the propensity scores an IPTW was created for each PLHIV and this weight was used to make a synthetic sample in which IPT and non-IPT groups were Balanced. The weighted sample was then used to determine the TB incidence rates and risk factors for developing TB disease. The weighted dataset was used for the survival analysis using TB disease diagnosis as the failure variable. Entry time was the first date seen in the CTC (after 1st January 2012) and exit time out was TB diagnosis date, or the last clinic visit recorded. TB incidence rates and corresponding 95% Confidence Intervals were calculated by dividing the sum of all TB episodes occurring during follow up and total time contributed by each of the study participant. For individuals who received IPT the time before receipt of IPT was considered "not on IPT" and the time following initiation of IPT considered "on IPT". Risk factors for developing TB disease among PLHIV were analyzed using regression model. The univariate Cox-regression model contained covariates either known to influence TB diagnosis from previous studies or from clinical experience of the authors. Multivariate regression included covariates with P-values less than 0.2 to obtain adjusted Hazard Ratios (HR) and 95% confidence intervals (95% CI). Multivariate analysis was done to control for possible residual confounding despite IPTW [30]. Cluster effect at the health facility level was checked in the regression model and were not significant. Kaplan Meier graphs for the effect of IPT on TB incidence were drawn to show comparison of TB disease probabilities between those who received IPT and those who did not receive IPT. Stata version 14 (College Station, TX: Stata Corp LP) was used for analysis.

## Ethical consideration

To maintain participants' confidentiality. the study used de-identified routinely collected HIV data. The data was fully de-identified before made accessible. Ethical review and approval was granted by Kilimanjaro Christian Medical College (KCMUCo) Institutional Review Board (IRB) which granted the ethical clearance certificate number 2287. The **parent** project known

as SEARCH was given ethical approval by NIMR with approval number National Institute of Medical Research (NIMR)/HQ/R.8c/Vol.II/961. The submitted manuscript is part of Doctor of Philosophy work within the main project. As the study used de-identified secondary data from routine clinical visits, the need for informed consent from the study participants was waived by both KCMUCo IRB and NIMR ethical approval. Data used for the study can be available for public use in the research repository of London School of Hygiene and Tropical Medicine, United Kingdom. The data can be accessed using the following link: https://www.lshtm.ac.uk/sites/default/files/201706/Tanzania%20CTC%20Documentation.pdf.

## Results

### (1) Baseline characteristics of study participants

A total of 171,743 PLHIV enrolled in 315 health facilities in Dar es Salaam, Iringa and Njombe regions from January 2012-December 2016. A total of 10,326 (6.01%) study participants were excluded from the analysis as they either had TB disease before CTC enrolment or had TB diagnosed at CTC before follow up or they were from clinics which never implemented IPT throughout the study period. Of the 161,417 who were involved in the analysis, only 1.00% of PLHIV developed TB disease among those who received IPT and 4.98% among those who never received IPT (Fig 1).

Majority of individuals who received IPT were females (71.60%), aged 25–49 years (77.19%), with working functional status (97.51%), in WHO clinical stage III (33.61%) not on ART (72.11%), in individuals with normal BMI (57.76%) and in PLHIV with normal nutritional status (93.81%). IPT initiation was also higher among those enrolled in 2014 (22.75%) Dar es Salaam region (74.72%, enrolled in hospital (49.27%) and enrolled in public health facilities (83.88%) (Table 1).

### (2) Balance of covariates after IPTW

After IPTW, the standardized differences for the weighted sample were closer to zero compared to the unweighted sample. The variances of the different covariates were closer to 1 compared to the variance in the unweighted sample.
(Table 2).

### (3) TB cases, follow up time and TB incidence rate

During follow up, a total of 7,052 new TB cases were diagnosed with a total of 278,545.90 person years giving an overall TB incidence rate of 25.32 (95% CI: 21.43–30.31) per 1,000 person-years. Using the weighted sample, an overall TB incidence of 11.57 (95% Confidence Interval (CI): 11.09–12.07) per 1,000 person-years was obtained. PLHIV who received IPT had lower TB incidence rate compared with those who never received IPT: 10.49 (95% CI: 9.11–12.15) per1,000 person-years versus 12.00 per 1,000 (95% CI: 11.69–12.33) per 1,000 person-years respectively. TB incidence rate was also lower higher among females; 9.27(95% CI: 8.76–9.82 per 1,000 person-years compared to males, in PLHIV aged 20–24 years: 6.10 (95% CI: 4.92–7.66) compared to other age groups, in PLHIV with working functional status; 11.27 (95% CI: 10.78–11.78) per 1,000 person-years than in other functional statuses and in those who were on ART: 8.38 (95% CI: 7.45–9.47) per 1,000 person-years compared with those not on ART. TB incidence was also decreased in those with WHO clinical stage I; 6.05 (95% CI:5.34–6.82) per 1,000 person-years compared to higher WHO clinical stages in those who were obese: 4.71 (95% CI: 3.59–6.32) per 1,000 person-years compared to those with other weights, in those with normal nutritional status: 10.83 (95% CI: 10.32–11.37) per 1,000 person-years compared

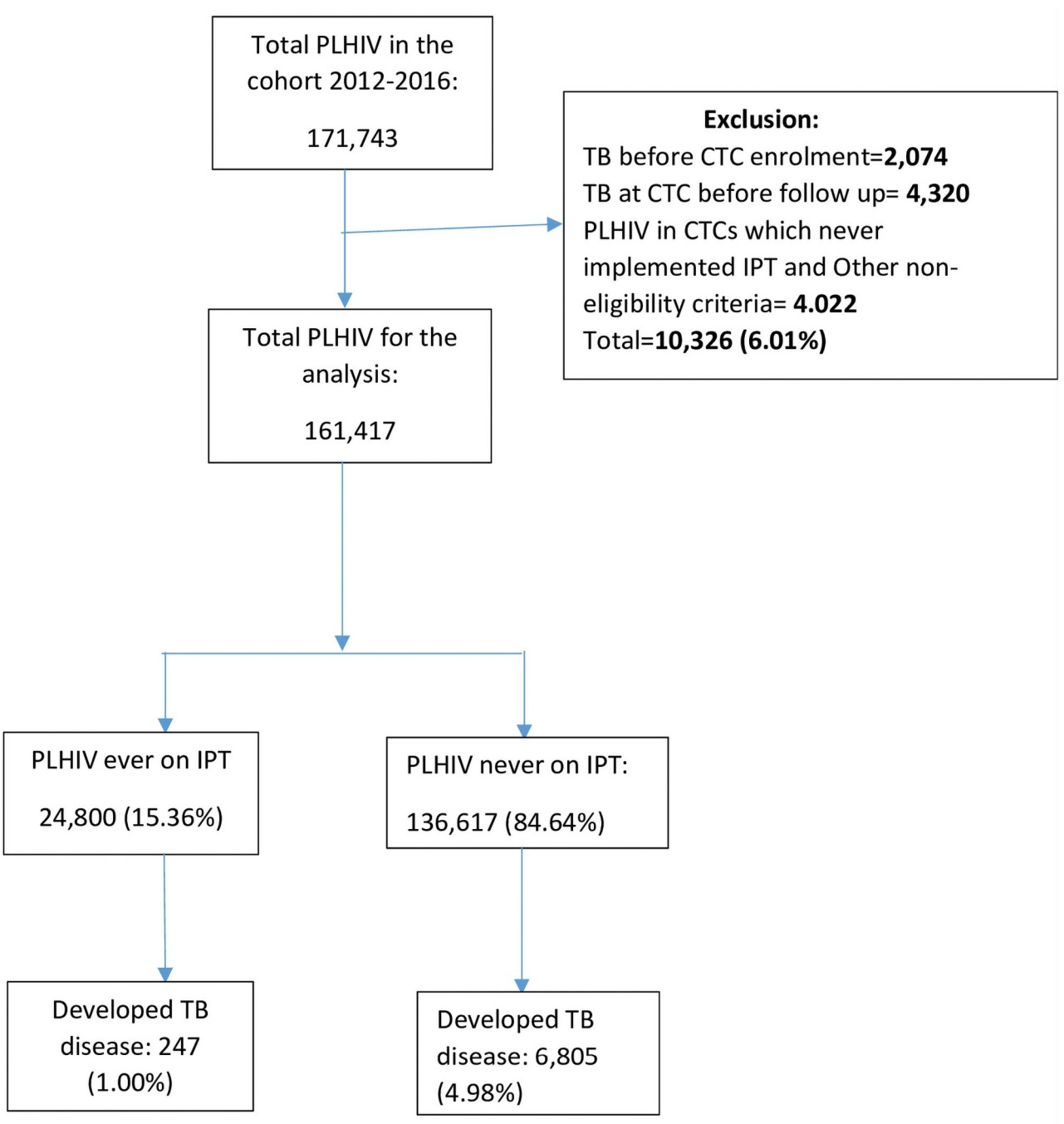

**Fig 1. Flow diagram for Isoniazid Preventive Therapy and TB diagnosis among PLHIV.** The figure showed only 15.36% of the PLHIV in the study were initiated on Isoniazid Preventive Therapy. When compared, less PLHIV who ever used IPT developed TB disease (1.00%) than those who did not use IPT (4.98%).

to other groups. TB incidence rate was also decreased in PLHIV enrolled in dispensaries: 10.35 (95% CI: 9.44–11.37) per 1,000 person-years compared to those in other health facility levels, in PLHIV enrolled in care in 2012: 9.20 (95% CI: 8.42–10.06) per 1,000 person-years compared to those in the following years, in those enrolled in Njombe region: 6.45 (95% CI: 5.77–7.25)

**Table 1. Baseline characteristics of study participants, N = 161,417.**

| Variable | IPT status | | | |
|---|---|---|---|---|
| | Never on IPT n = 136,617 | Ever on IPT n = 24,800 | Total | P-value |
| **Sex** | | | | |
| Male | 42,409 (31.04%) | 7,044 (28.40%) | 49,453 (30.63%) | P<0.001 |
| Female | 94,208 (68.96%) | 17,756 (71.60%) | 111,964 (69.36%) | |
| **Baseline age** | Mean: 33.64 years, SD = 12.44 years, Range: 0–95 years | | | |
| 0–9 | 6,741(4.93%) | 567 (2.29%) | 7,308 (4.53%) | P<0.001 |
| 10–19 | 6,458 (4.73%) | 739 (2.98%) | 7,197 (4.46%) | |
| 20–24 | 13,977 (10.23%) | 1,808 (7.29%) | 15.785 (9.78%) | |
| 25–49 | 96,848 (70.89%) | 19,143 (77.19%) | 115, 991 (71.86%) | |
| + 50 | 12,593 (9.22%) | 2,543 (10.25%) | 15,136 (9.38%) | |
| **Baseline functional status** | | | | |
| Ambulatory | 4,222 (3.10%) | 475 (1.92%) | 4,697 (2.92%) | P<0.001 |
| Bedridden | 1,143 (0.84%) | 139 (0.56%) | 1,282 (0.80%) | |
| Working | 130,642 (96.06%) | 24,069 (97.51%) | 154,711 (96.28%) | |
| **Baseline WHO clinical stage** | | | | |
| I | 52,176 38.70% | 8,175 (33.39%) | 60,351 (37.89%) | P<0.001 |
| II | 32,513 24.12% | 6,684 (27.30%) | 39,197 (24.61%) | |
| III | 40,051 29.71% | 8,230 (33.61%) | 48,281 (30.21%) | |
| IV | 10,073 7.47% | 1,398 (5. 71%) | 11,471 (7.20%) | |
| **Baseline ART status** | | | | |
| No ART | 85,297 (62.90%) | 17,749 (72.11%) | 103,046 (64.31%) | P<0.001 |
| ART | 50,321 (37.10%) | 6,865 (27.89%) | 57,186 (35.69%) | |
| **Baseline BMI** | | | | |
| Underweight | 14,806 (17.13%) | 2,584 (14.18%) | 17,390 (16.61%) | P<0.001 |
| Normal | 48,697 (56.33%) | 10,524 (57.76%) | 59,221 (6.57%) | |
| Overweight | 15,598 (18.04%) | 3,465 (19.02%) | 19,063 (18.21%) | |
| Obesity | 7,356 (8.51%) | 1,648 (9.04%) | 9,004 (8.60%) | |
| **Baseline nutritional status** | | | | |
| Normal | 120,051 (92.35%) | 22,438 (93.81%) | 142,489 (92.58%) | P<0.001 |
| Moderate | 8,501 (6.54%) | 1,335 (5.58%) | 9,836 (6.39%) | |
| Severe | 1,445 (1.11%) | 146 (0.61%) | 1,591 (1.03%) | |
| **CTC enrolment year** | | | | |
| 2012 | 24,364 (17.83%) | 4,378 (17.65%) | 28,742 (17.81%) | P<0.001 |
| 2013 | 26,603 (19.47%) | 5,017 (20.23%) | 31.620 (19.59%) | |
| 2014 | 28,491 (20.85%) | 5,643 (22.75%) | 34,134 (21.15%) | |
| 2015 | 26,215 (19,19%) | 5,031 (20.29%) | 31,246 (19.36%) | |
| 2016 | 30,944 (22.65%) | 4,731 (19.08%) | 35,675 (22.10%) | |
| **Region at enrolment** | | | | |
| Dar es Salaam | 96,524 (70.65%) | 18,556 (74.82%) | 115,080 (71.29%) | P<0.001 |
| Iringa | 17,216 (12.60%) | 1,238 (4.99%) | 18.454 (11.43%) | |
| Njombe | 22,877 (16.75%) | 5,006 (20.19%) | 27,883 (17.27%) | |
| **Health Facility Type at enrolment** | | | | |
| Dispensary | 55,932 (40.94%) | 6,232 (25.13%) | 62,164 (38.51%) | P<0.001 |
| Health Center | 36,619 (26.80%) | 6,348 (25.60%) | 42,967 (26.62%) | |
| Hospital | 44,066 (32.26%) | 12,220 (49.27%) | 56,286 (34.87%) | |
| **Health Facility ownership at enrolment** | | | | |

(*Continued*)

**Table 1.** (Continued)

| Variable | IPT status | | | |
| --- | --- | --- | --- | --- |
| | Never on IPT n = 136,617 | Ever on IPT n = 24,800 | Total | P-value |
| Public | 96,211 (70.42%) | 20,803 (83.88%) | 117,014 (72.49%) | P<0.001 |
| Private | 40,406 (29.58%) | 3,997 (16.12%) | 44.403 (27.51%) | |

95% CI = 95% Confidence Interval, ART = Antiretroviral Treatment, BMI-Body Mass Index, Care and Treatment Clinic, IPT = Isoniazid Preventive Therapy, TB = Tuberculosis, SD = Standard Deviation, WHO = World Health Organization.

per 1,000 person-years compared to those registered in other regions and in those enrolled in private health facilities: 10.37 (5% CI:9.24–11.67–10.29) per 1,000 person-years compared to those enrolled in public health facilities (Table 3).

Kaplan-Meier graph comparing TB disease between those who received IPT and those who did not use IPT was fitted. With increasing time of follow up cumulative incidence of TB disease increased in both groups. However, the incidence was higher among those who did not receive IPT compared with those who received the intervention (Fig 2).

### (4) Risk factors for TB disease

In multivariate analysis of unweighted sample, the risk of developing TB disease was less by 13% among those who ever received IPT: adjusted Hazard Ratio (aHR) = 0.87 (95% CI:0.76–1.01; P>0.05) compared to those who never used IPT. Using weighted sample, after adjusting for other covariates the risk of TB incidence was reduced by 52% in the IPT group compared to non-IPT; aHR = 0.48 (95% CI: 0.40–0.58; P<0.001). In the weighted sample, the risk of developing TB disease was also lower among females: aHR = 0.55 (95% CI: 0.50–0.61), P<0.001, in those who were obese: aHR = 0.36 (0.26–0.48), P<0.05, in those on ART: aHR = 0.61 (95% CI:0.53–0.70), P<0.001, and in those enrolled in Njombe region: aHR = 0.28 (95% CI:0.23–0.33), P<0.001. The risk of TB disease was increased in individual with WHO clinical stage IV; aHR = 3.44 (95% CI: 2.84–4.17), P<0.001 (Table 4).

## Discussion

We set out to determine impact of IPT on TB incidence among PLHIV as well show case the application of IPTW to minimize selection bias in observational studies in order to

**Table 2. Balance between unweighted and weighted samples after Inverse Probability of Treatment Weighting.**

| Covariate | Standardized Difference | | Variance | |
| --- | --- | --- | --- | --- |
| | Unweighted sample | Weighted sample | Unweighted sample | Weighted sample |
| **Sex** | 0.03721 | -0.01170 | 0. 9636 | 1.0162 |
| **Baseline age** | 0.1489 | 0.0108 | 0.7395 | 0.8963 |
| **Baseline functional status** | 0.8073 | -0.0034 | 0.5981 | 1.0154 |
| **Baseline ART status** | -0.2385 | -0.0019 | 0.8344 | 0.9989 |
| **Baseline BMI** | 0.0559 | -0.0130 | 0.9685 | 0.9502 |
| **Baseline nutritional status** | -0.0736 | 0.0003 | 0.7400 | 0.9962 |
| **Health Facility type at enrolment** | 0.3583 | 0.0056 | 0.9118 | 0. 9695 |
| **Region at enrolment** | 0.0147 | 0.0123 | 1.2120 | 1.1355 |
| **Health Facility ownership at enrolment** | 0.2935 | -0.0134 | 0.5677 | 1.0206 |

ART = Antiretroviral Therapy, BMI-Body Mass Index.

**Table 3. The incidence of TB cases, total person-year and TB rates among 24,800 PLHIV who received IPT and a weighted sample of those who did not receive IPT.**

| Variable | Number of TB cases | Total person-years (Years) | Rate/1,000 (95% CI) |
|---|---|---|---|
| **Overall** | 1,037 | 89,661 | 11.57 (11.09–12.07) |
| **IPT status** | | | |
| Never on IPT | 765 | 63,739 | 12.00 (11.69–12.33) |
| **Ever on IPT** | **272** | **25,922** | **10.49 (9.11–12.15)** |
| **Sex** | | | |
| Male | 450 | 26,335 | 17.10 (16.05–18.23) |
| **Female** | **587** | **63,327** | **9.27 (8.76–9.82)** |
| **Baseline age** | | | |
| 0–19 | 105 | 7,442 | 14.17 (11.68–17.37) |
| **20–24** | **45** | **7,427** | **6.10 (4.92–7.66)** |
| 25–49 | 764 | 66,226 | 11.54 (11.03–12.07) |
| +50 | 122 | 8,566 | 14.27 (12.69–16.11) |
| **Baseline functional status** | | | |
| Ambulatory | 54 | 2,545 | 21.06 (18.25–24.35) |
| Bedridden | 9 | 606 | 14.68 (9.72–23.05) |
| **Working** | **975** | **86,511** | **11.27 (10.78–11.78)** |
| **ART status** | | | |
| No on ART | 824 | 64,279 | 12.82 (12.28–13.40) |
| **ART initiated** | **209** | **24,937** | **8.38 (7.45–9.47)** |
| **Baseline WHO clinical stage** | | | |
| **Stage I** | **194** | **32,060** | **6.05 (5.34–6.82)** |
| Stage II | 232 | 24,647 | 6.40 (8.58–10.39) |
| Stage III | 485 | 26,317 | 18.43 (17.54–19.39) |
| Stage IV | 119 | 5,678 | 21.91 (17.99–24.43) |
| **Baseline Body Mass Index** | | | |
| Underweight | 208 | 9,349 | 20.25 (20.81–23.81) |
| Normal | 398 | 33,280 | 11.95 (11.29–12.36) |
| Overweight | 75 | 10,888 | 6.84 (5.94–7.92) |
| **Obesity** | **25** | **5,221** | **4.71 (3.59–6.32)** |
| **Baseline nutritional status** | | | |
| **Normal** | **859** | **79,304** | **10.83(10.32–11.37)** |
| Moderate | 116 | 5,270 | 21.95 (19.99–24.14) |
| Severe | 23 | 784 | 29.59 (22.58–39.03) |
| **Health Facility type at enrolment** | | | |
| **Dispensary** | **338** | **32,666** | **10.35 (9.44–11.37)** |
| Health center | 253 | 23,661 | 10.70 (9.95–11.02) |
| Hospital | 446 | 33,334 | 13.38 (12.66–14.14) |
| **CTC enrolment year** | | | |
| **2012** | **227** | **24,631** | **9.20 (8.42–10.06)** |
| 2013 | 230 | 23,720 | 9.69 (8.89–10.60) |
| 2014 | 216 | 20,767 | 10.42 (9.55–11.40) |
| 2015 | 188 | 13,014 | 14.45 (13.04–16.06) |
| 2016 | 176 | 7,530 | 23.38 (21.01–26.10) |
| **Region at enrolment** | | | |
| Dar es Salaam | 850 | 62,545 | 13.60 (13.00–14.23) |
| Iringa | 78 | 10,323 | 7.58 (6.08–9.57) |

*(Continued)*

**Table 3.** (Continued)

| Variable | Number of TB cases | Total person-years (Years) | Rate/1,000 (95% CI) |
|---|---|---|---|
| **Njombe** | **108** | **16,793** | **6.45 (5.77–7.25)** |
| **Health Facility ownership at enrolment** | | | |
| **Private** | **274** | **26,420** | **10.37 (9.24–11.67)** |
| Public | 763 | 63,241 | 12.07 (11.75–12.79) |

95% CI = 95% Confidence Interval, ART = Antiretroviral Treatment, CTC = Care and Treatment Clinic,

IPT = Isoniazid Preventive Therapy, TB = Tuberculosis, WHO = World Health Organization.

establish causal relationship between IPT and TB incidence among PLHIV in routine clinical settings.

These results show that in the implementation of IPT the risk of developing TB Disease was reduced by a half (52%) in those who received IPT in the studied cohort when compared to those who never received IPT after applying IPTW and adjusting for confounding from other factors. The results also show lower TB disease incidence among those who received IPT in comparison with those who never received IPT. Both lower rates of TB among those who ever used IPT and reduced TB risk among IPT users consolidates evidence of effectiveness of IPT on TB incidence among PLHIV. Lower TB incidence among IPT users has also been documented in other studies where PLHIV who ever used IPT had lower rates of TB incidence than those who never used IPT [4, 18, 31–34], however, unlike many of these studies in real

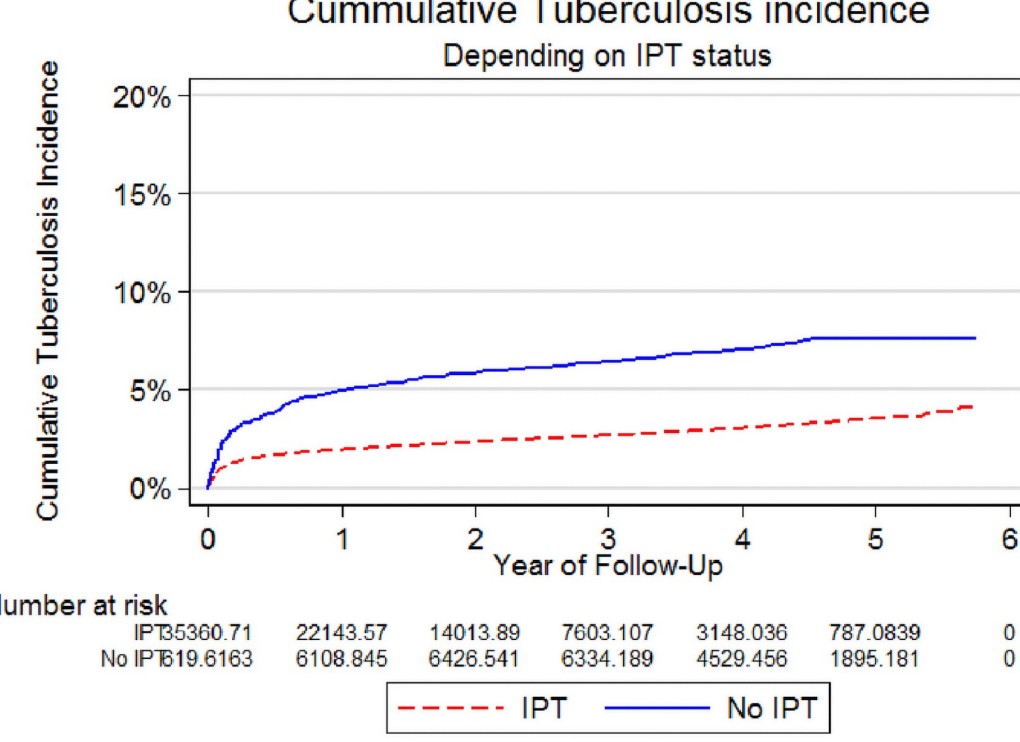

**Fig 2. Kaplan Meier graph showing cumulative Tuberculosis incidence depending on the status of Isoniazid Preventive Therapy.** Throughout the follow up, those who ever received IPT had less cumulative Tuberculosis incidence than those who did not receive the intervention.

**Table 4. Risk factors for TB disease using weighted sample.**

| Variable | Univariate | | Multivariate | |
|---|---|---|---|---|
| | cHR, 95% CI | P-Value | aHR, 95% CI | P-Value |
| **IPT status** | | | | |
| Never on IPT | 1 | P>0.05 | 1 | P<0.001 |
| Ever on IPT | 0.87 (0.76–1.01) | | 0.48 (0.40–0.58) | |
| **Sex** | | | | |
| Male | 1 | P<0.001 | 1 | P<0.001 |
| Female | 0.54 (0.50–0.53) | | 0.55 (0.50–0.61) | |
| **Baseline age** | | | | |
| 0–19 | 1 | P>0.05 | | |
| 20–24 | 0.43 (0.32–0.58) | | | |
| 25–49 | 0.81 (0.67–0.99) | | | |
| +50 | 1.00 (0.80–1.27 | | | |
| **Baseline BMI** | | | | |
| Underweight | 1 | P<0.001 | 1 | P<0.001 |
| Normal | 0.54 (0.49–0.59) | | 0.71 (0.64–0.79) | |
| Overweight | 0.31 (0.26–0.36) | | 0.46 (0.38–0.54) | |
| Obesity | 0.21 (0.16–0.28) | | 0.36 (0.26–0.48) | |
| **Baseline functional status** | | | | |
| Ambulatory | 1 | P>0.05 | | |
| Bedridden | 0.70 (0.45–1.08) | | | |
| Working | 0.54 (0.46–0.63) | | | |
| **Baseline ART status** | | | | |
| ART not initiated | 1 | P<0.001 | 1 | P<0.001 |
| ART initiated | 0.65 (0.58–0.74) | | 0.61 (0.53–0.70) | |
| **Baseline WHO clinical stage** | | | | |
| Stage one | 1 | P<0.001 | 1 | P<0.001 |
| Stage two | 1.55 (1.33–1.81) | | 1.51 (1.28–1.78) | |
| Stage three | 3.04 (2.68–3.46) | | 3.03 (2.64–3.48) | |
| Stage four | 3.45 (2.85–4.18) | | 3.44 (2.84–4.17) | |
| **Nutritional status** | | | | |
| Normal | 1 | | 1 | |
| Moderate | 2.03 (1.82–2.26) | P<0.001 | 1.19 (1.05–1.35) | P>0.05 |
| Severe | 2.73 (2.05–3.65) | | 1.22 (0.87–1.72) | |
| **Health Facility type** | | | | |
| Dispensary | 1 | P>0.05 | | |
| Health center | 0.98 (0.86–1.11) | | | |
| Hospital | 1.14 (1.00–1.28) | | | |
| **CTC enrolment Year** | | | | |
| 2012 | 1 | | | |
| 2013 | 1.05 (0.93–1.19) | P>0.05 | | |
| 2014 | 1.13 (1.00–1.29) | | | |
| 2015 | 1.57 (1.37–1.80) | | | |
| 2016 | 2.54 (2.21–2.93) | | | |
| **Region** | | | | |
| Dar es Salaam | 1 | | 1 | P<0.001 |
| Iringa | 0.56 (0.44–0.70) | P<0.001 | 0.33 (0.25–0.42) | |
| Njombe | 0.47 (0.42–0.54) | | 0.28 (0.23–0.33) | |

(*Continued*)

**Table 4.** (Continued)

| Variable | Univariate | | Multivariate | |
|---|---|---|---|---|
| | cHR, 95% CI | P-Value | aHR, 95% CI | P-Value |
| **Health facility ownership** | | | | |
| Private | 1 | | 1 | P>0.05 |
| Public | 1.16 (1.03–1.32) | P<0.001 | 0.89 (0.75–1.11) | |

95% CI = 95% Confidence Interval, aHR = Adjusted Hazard Ratio, ART = Antiretroviral Treatment, BMI-Body Mass Index, Care and Treatment Clinic, cHR = Crude Hazard Ratio, CTC = Care and Treatment Clinic, IPT = Isoniazid Preventive Therapy, TB = Tuberculosis, WHO = World Health Organization.

life, the current study applied IPTW to minimize bias associated with non-experimental study designs. Protective effect of IPT on TB incidence among PLHIV in non-research, routine HIV care settings have also been determined in other studies when those who used IPT were compared with those who did not use in routine settings in Tanzania [20], Ethiopia [4, 18, 34–36] and a systematic review [16], and it ranged from 48% to 94%. The various studies showed that the protective effect of IPT is stretched over a long range. These studies were conducted in different routine settings with varying degrees of data quality together with different approaches in controlling for possible biases during data analysis level. Hence, the range of IPT effectiveness in these studies could be due to different study settings and different statistical manipulations done to make those who received IPT and those who did not receive comparable groups.

Overall this population had a relatively lower TB incidence reduction compared to other similar studies in Sub-Saharan Africa including the study in Tanzania [20]. The discrepancy between our study and other studies could also be due to lack of application of statistical techniques like propensity scores to minimize selection bias associated with observational studies [23, 29]. Lack of use of such techniques to minimize selection bias in establishing causal inference in observation studies may result in erroneous estimation of the outcome of interest [37].

The results from the analysis of the weighted data also showed that the risk of TB disease was also lower among females, those on ART, obese individuals and those enrolled in Njombe region. The risk of TB disease was higher in PLHIV with baseline WHO clinical stage IV. Other studies have also shown a low TB risk among females in comparison with Males [38, 39]. Difference in health seeking behavior between males and females [38] and the role of biological difference between the two sexes playing a role in immune responses in resisting *Mycobacterium Tuberculosis* have been attributed to the differences [39]. ART use is a known public health tool in reducing the risk of developing TB disease among PLHIV [2, 40, 41]. ART restores immune responses to *Mycobacterium Tuberculosis* through lowering of HIV viral load and hence, prevents HIV-associated TB [41]. Lower TB disease risk among individuals with higher BMI (overweight and obese) in comparison with normal nutritional status has also been reported by others [42, 43]. Overweight or obese people may have increased intake of protein and energy containing food as well as micronutrients which may be protective of TB diseases [43]. However, a thorough mechanism of the association between nutritional status and TB disease may be needed. Different TB risk according to regions could be attributed to health system components which were beyond the scope of this study. Regional variations in TB disease in Tanzania has also been reported by others [44]. The risk of TB disease in individuals with advanced WHO clinical stages (stage 3 and 4) is also reported in other studies [45, 46]. Advanced HIV affects both cellular and hormonal immune responses required to control TB, thus increasing TB disease susceptibility in individuals with advanced HIV disease [47].

ICF and IPT in Tanzania were scaled up from 2011. The observed increased TB notification with increasing years from 2012 could be due to scaling up of these interventions which result

into increased TB case notification [48]. TB diagnosis especially by using Florescent Light Emitting Diode (LED) microscopy and Gene-Xpert has been improved over time. Use of these diagnostic tests which perform better than conventional light microscopy lead to increased TB case notification [49, 50]. Moreover, the fact that advancement in TB diagnosis happened as years go, this may explain the increased TB notification from 2012 to 2016. Differences in TB burden among regions in Tanzania was also documented by others [51]. Different TB burden among regions together with differences in TB diagnosis among regions in the country may explain different TB incidence found in this study. Lower TB notification in private health facilities reported in the current research is similar to those reported by others [52]. This may be due to less engagement of private health facilities in TB disease management [53].

This study has an implication on practice and policy in preventing TB. The findings of this study have added information on how IPT works in reducing TB diseases among PLHIV in routine real-world settings. Such findings encourage policy makers to continue safeguarding policies on TB preventive strategy. The findings also encourage health care providers to continue scaling up IPT such that it is accessed by all eligible PLHIV in the country.

The strength of this analysis is that the data are taken from routine visits by PLHIV attending CTC and hence the results represent the effect of IPT under routine settings. The data came from three regions representing a cross section of Tanzania where the prevalence was high [25]. The analysis used IPTW to adjust for the characteristics of study participants [23].

The main limitation of the study as a result of its being a retrospective design, is its inability to include other variables which could be included as possible confounders for IPT initiation. These are such as availability of Isoniazid tablets in the health facilities which may affect IPT initiation. Another limitation of the study is its inability to complete quantitative data with qualitative data to understand context of IPT delivery. Moreover, the study may result into overestimating TB among non-IPT recipients because much TB may have been diagnosed more at the beginning of follow up.

## Conclusion

The study found that IPT reduced TB disease in routine HIV care and treatment settings in Tanzania to the same extent as other studies in routine settings. This study has also shown the possibility of applying IPTW to determine causality in observational studies.

## Supporting information

**S1 Dataset.**
(ZIP)

## Acknowledgments

We would like to acknowledge active participation of health care providers and clinic clients in the participating health facilities for their involvement in the study. The authors would also like to thank staff from National AIDS Control Program Tanzania for their support throughout the study.

## Author Contributions

**Conceptualization:** Werner M. Maokola.

**Formal analysis:** Werner M. Maokola, Michael J. Mahande.

**Funding acquisition:** Werner M. Maokola, Jim Todd.

**Investigation:** Werner M. Maokola.

**Methodology:** Werner M. Maokola, Michael J. Mahande.

**Project administration:** Werner M. Maokola.

**Software:** Werner M. Maokola, Jim Todd.

**Supervision:** Werner M. Maokola, Bernard J. Ngowi, Jim Todd, Sia E. Msuya.

**Validation:** Masanja Robert.

**Visualization:** Jim Todd, Masanja Robert.

**Writing – original draft:** Werner M. Maokola.

**Writing – review & editing:** Bernard J. Ngowi, Michael J. Mahande, Jim Todd, Masanja Robert, Sia E. Msuya.

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
