## [Decision Letter · Decision Letter 0]

20 Jan 2021

PONE-D-20-37236

Impact of Isoniazid Preventive Therapy on Tuberculosis incidence among people living with HIV; a secondary data analysis using Inverse Probability Weighting of patients attending HIV care and treatment clinics in Tanzania.

PLOS ONE

Dear Dr. Maokola,

Thank you for submitting your manuscript to PLOS ONE. After careful consideration, we feel that it has merit but does not fully meet PLOS ONE’s publication criteria as it currently stands. Therefore, we invite you to submit a revised version of the manuscript that addresses the points raised during the review process.

We look forward to receiving your revised manuscript.

Kind regards,

Katalin Andrea Wilkinson, PhD

Academic Editor

PLOS ONE

Journal Requirements:

2. Please include additional information regarding the data extraction tool used in the study and ensure that you have provided sufficient details that others could replicate the analyses. For instance, if you developed a data extraction tool as part of this study and it is not under a copyright more restrictive than CC-BY, please include a copy, in both the original language and English, as Supporting Information, or include a citation if it has been published previously.

"The study was funded through SEARCH Project under Bill and Melinda Gates Foundation. Routine HIV data management is Tanzania is managed collaboratively by the Government of the Republic of Tanzania and the Government of the United States of America through President’s Emergency Plan for AIDS Relief."

Reviewers' comments:

Reviewer's Responses to Questions

**Comments to the Author**

1. Is the manuscript technically sound, and do the data support the conclusions?

Reviewer #1: Partly

Reviewer #2: Yes

Reviewer #3: Yes

Reviewer #4: Yes

2. Has the statistical analysis been performed appropriately and rigorously? 

Reviewer #1: No

Reviewer #2: Yes

Reviewer #3: Yes

Reviewer #4: Yes

3. Have the authors made all data underlying the findings in their manuscript fully available?

Reviewer #1: No

Reviewer #2: No

Reviewer #3: Yes

Reviewer #4: No

4. Is the manuscript presented in an intelligible fashion and written in standard English?

Reviewer #1: Yes

Reviewer #2: Yes

Reviewer #3: No

Reviewer #4: Yes

5. Review Comments to the Author

Reviewer #1: In this manuscript, the authors make use of a national dataset to explore the effects of IPT on incident TB among people living with HIV in Tanzania. To address confounding in such an observational dataset, they use an inverse probability of treatment weighting to balance certain confounders between treated and untreated groups. The large size of the dataset and the appropriately chosen method for causal inference are promising, but there are several weaknesses of the analysis as currently presented:

1. The 4 variables adjusted for are not sufficient to balance the IPT and non-IPT groups. IPTW requires balancing all confounders that independently influence both IPT initiation and TB risk. In this dataset, there are a number of other measured but unadjusted confounders that appear to have been associated with IPT initiation (Table 1) and that are also likely to influence TB diagnosis outside of the IPT causal pathway. These include, for example, ART status, BMI and/or nutritional status, and type of health facility. The IPTW analysis is not valid unless all confounders are both measured and included in the weighting, and the authors cannot just assert that “after weighting, the two exposure groups were comparable.”

The authors also note in the introduction that IPT tended to be introduced sooner at larger facilities, and larger facilities may also be associated with differences in care that influence either risk of developing TB or probability of being diagnosed with TB (e.g. because of different TB diagnostics available), making this another potential confounder.

2. The authors need to do more to describe the motivation and context for their study. While Plos One does not select papers for publication based on “impact”, it is still important to accurately describe the existing literature and what the current analysis adds. The authors start from the assertion that “IPT reduces TB incidence” (abstract), and they cite a number of observational studies demonstrating effectiveness of IPT in reducing TB incidence in routine care settings including Tanzania. They say that the current study will help to inform scale-up, but it is not clear what aspects of scale-up this study informs. Is there clinically important uncertainty in the size of effect based on prior literature? Is there something different about this setting that might lead to a different effect? Are there particular covariates of interest that this analysis is intended to explore?

Similarly, it seems contradictory in the introduction to state that “a number of studies … in routine public health settings… have also shown benefits of IPT on TB incidence ”,but then assert in the next paragraph that “effectiveness [of IPT] in these settings has not been adequately and properly evaluated.” More exploration is needed of what the gaps in current data are and how this study addresses them.

3. I note in Figure 2 that a lot of the divergence between the IPT and no-IPT groups occurs within the first two months of follow up. This makes me think that much of the difference is not an effect of the IPT, but rather a difference in the suspicion of existing TB: Clinicians may have been disinclined to start IPT when they suspected that a patient already had active TB, even if an initial diagnostic test was negative or they were still waiting for a diagnostic result. Or, it could be that patients who were going to start IPT got a more thorough evaluation to rule out active TB at baseline, making those with early, undiagnosed TB disease less likely to be included in the IPT group than in the no-IPT group. More information about TB screening and diagnostic practices during the study period might help to clarify this, and it is also a limitation that needs to be explored in the discussion.

Minor:

Introduction:

- It’s potentially misleading to cite the % of PLHIV who have TB infection at time of death. Would be more useful to cite the % of PLHIV who have TB disease at time of death.

Methods:

- Please state how incidence rates (and their confidence intervals) were calculated.

- How was the multivariable model (Table 3) developed? Why this set of covariates?

Results:

- Too many significant digits are included, making results both overly precise and difficult to read.

- Person-years do not make sense as reported. Should these be thousands of person years?

- What is the difference between “ambulatory” and “walking” functional status? To me, these seem synonymous.

- Sentences containing lists are difficult to follow due to inconsistent grammar and punctuation.

- Several numerical results seem to be incorrectly typed. For example, incidences by ART status differ between text and table, some confidence intervals are concerningly asymmetric, some column percentages in tables do not add to 100%, and some reported p values seem incorrect based on the point estimates and Cis presented.

- TB incidence seems to increase over time from 2012 to 2016. What there a change in diagnostic practice during this period (e.g. introduction of Xpert or change in screening practices), and if so, how is that likely to affect your results? Should calendar year or clinic-level diagnostic availability be adjusted for?

Discussion:

- I suggest presenting results in the context of follow-up period. IPT reduced TB incidence by xx% over what median duration of follow up? Effects are likely to wane with time, as reinfection increasingly predominates over reactivation as the source of incident TB.

- The main limitation cited is the “inability to include other [confounding] variables”, but they in fact have data on many more potentially-confounding variables that they do have the ability to include if they choose.

- Much more exploration of the study’s limitations is needed.

Author contributions:

- All authors should review and approve the final manuscript.

Reviewer #2: Thank you very much for allowing me to review this interesting manuscript.

The manuscript adds to the existing evidence on the effectiveness of IPT in reducing TB incidence based on analysis of a large number of participants in Tanzania. The papers showed durable protection against TB for 6 years. The manuscript is generally well written but lacks some details. Please consider suggestions below to improve clarity.

1. The definition of the IPT group and the control group is not clear to me. The authors conducted the survival analysis by defining entry time as the data of enrolment. Were participants classified into either IPT group or the control group depending on whether they started IPT at the time of enrolment? I presume that most of PLHIV started at IPT the time enrolment but some might have initiated it later (e.g. after investigation for TB). How did you account for those who started IPT later? For example, if they started IPT one month later, were they then transferred into the IPT group? If they remained in the control group, then the control group is not “those who never received IPT”.

2. What is the definition of TB disease? Only bacteriologically confirmed? Does this include TB diagnosed regardless of the timing after the follow-up? (e.g. TB diagnosed within one week after the enrolment.)

3. Figure 2 shows a sharp increase in TB very early in the follow-up. If I understand correctly, participants who screened positive didn’t start immediately upon enrolment and were included in the control group. Among those who screened positive, prevalent TB may have been found after investigation and inflated the number of TB cases in the control group in the initial period.

4. The authors claim that baseline characteristics were balanced after IPTW. However, I seem to find some imbalances, for example:

ART 29.02% vs 36.09%

Private facility 17% vs 29%

Underweight 15% vs 18%

Is this the reason why the authors conducted multivariable cox-regression after IPTW?

I wonder if the authors conducted balance diagnostics. Presenting results may be helpful. https://www.ncbi.nlm.nih.gov/pmc/articles/PMC4626409/

I would also suggest presenting the distribution of the weights.

Minor points

Introduction

Suggest mentioning the coverage of IPT among PLHIV not only the proportion of CTC providing TPT.

Also, suggest to briefly summarize the durability of efficacy reported in high TB burden countries and discuss how the present research adds to the evidence.

Method:

Is TST recommended at all?

If I remember correctly, the TB guidelines in Tanzania recommend repeating IPT after 2 years. Is there any chance that some of the participants repeated IPT?

Data collection

How did you assess nutritional status? What is the definition of each category?

How is ambulatory different from walking?

Do you have information on the previous history of TB?

Analysis

Please justly the use of p-values less than 0.05 as a criterion to select covariates included in the final model. This may miss important variables associated with exposure and outcome. Have you considered a higher threshold? https://pubmed.ncbi.nlm.nih.gov/8256780/

How did you handle missing data?

How were the weights given? Was it simple inverse or did you use stabilized weights?

Result

Paragraph 2

“from private health facilities (84.36%).” This is not consistent with Table 1. 84.36% is from public facilities.

Paragraph 3

Suggest reporting the median duration of follow-up in the two groups. The person-years reported don’t seem to correct. Only 134.56 person-year? Is this for all participants or 2309 TB patients? Either way, the number is too small, corresponding to a very short period of follow-up. Person-years in Table 2 are also too small.

Did you find any difference in the number of people who were lost to follow-up or died between the two groups? How could that impact the analysis?

Discussion

Paragraphs 2 and 3 seem repetitive.

The discussion could also mention the durability of protection.

Table 1.

Proportions of the weighted sample enrolled by year in the IPT group do not sum up to 100%.

Reviewer #3: There is no statement on the ethical consideration that explains how to keep the confidentiality of the participants.try to include a statement. It also needs some grammatical error corrections. both on the introduction and discussion.

Reviewer #4: ‘Impact of Isoniazid Preventive Therapy on Tuberculosis incidence among people living with HIV; a secondary data analysis using Inverse Probability Weighting of patients attending HIV care and treatment clinics in Tanzania.

This manuscript is intended to describe the use of inverse probability weighting (IPW) to determine the effectiveness of IPT in preventing active TB among PLHIV attending health facilities in Tanzania. The manuscript is good addition to scientific literature as it demonstrates the benefit of using IPW and its results. Previous studies on impact of TB prevention and TB Incidence in Tanzania has been done but the uniqueness of this paper is that it has included children, more regions in addition to Dar es Salaam and use of IPW to determine TB incidence among those on TB Prevention. The author needs to define and clearly state the objective of this study.

The abstract section is well written and reflecting on the manuscript. However, the author has indicated 171,672 as the study participants but this is not reflected in the results section which indicates 166,709 can this be clarified. The background section seems to be missing the objective and the statement on routine settings should be clear they should consider routine care setting. The conclusion should be aligned to the study objectives.

In the background section, in the 2nd paragraph the author has indicated Tanzania adopting WHO 3 I's in 2011 , can this be expounded was it in the form of guidelines or policy directive from the ministry ? and further expound on the scale up was it phased from higher level to lower level .The author has indicated > 50% scale up in the CTCs can this indicated in patient numbers as well .The policy on IPT uptake is it once in a lifetime or repeated after a certain duration of time in Tanzania .The fourth paragraph should read limited information on effectiveness of IPT on TB incidence as there has a study published on this (Sabasaba et al. BMC Infectious Diseases)

The manuscript would have benefitted from a clearer ,detailed and a logical methodology section for the readers to understand .The authors should consider to elaborate on the study design , clearly describe the study setting or a brief on the 3 regions .A brief explanation on the TB/HIV service delivery in particular IPT services in the health facilities in this 3 regions , the TB screening process , diagnosis , treatment and follow up .The authors needs to be clear on the study population .In the data analysis section the author needs to clearly indicate how the 4 variables ( Age,sex, region WHO staging ) was achieved at, justify why IPW .A clear explanation on the propensity scores used and how this was achieved .There should be a step wise approach with clear formulae and outcomes .

In the result section, the authors need to have a clearly structured sub section of the results. There needs to be clarification on the actual number of study participants analyzed (171,743 or 171,672 or 166,709?) and let it be uniform throughout the document. Table 1 is not uniform and a brief explanation on the age stratification from 0-9 then 10 – 19 then 20 – 24 then 25- 29. There needs to a clear write up of the highlight summary findings of the 3 tables in the result section and revision of the 3 tables as they appear overcrowded .

In the discussion section, in the first paragraph the aspect of using IPTW to determine effectiveness of public health intervention has been previously used refer to the NIH study (Sheri A. Lippman et al .NIH ) it’s more of use of TB prevention and in this country setting that is limited .The author seems to have duplicate information on ‘IPT lowering TB incidence by 70% ‘ in the 2nd and 4th paragraph can this be revised .From the commencement of TB prevention in the HIV facilities from 2011 has the national TB program seen some changes like reduction in the TB/HIV co-infection rate ? . Can the author expound on the association of higher TB incidence in the variables: Male; not on ART ; aging population ; Nutrition status ; higher numbers in 2016 and in the Dar es salaam region and the public health facilities of TB incidence .Basically the authors need to further expound on the findings of the current study in line with the results they got.

Finally, perhaps the authors can strengthen the conclusion in line with the objective of the study perhaps inclusion on use of IPW in evaluating effectiveness of a public health intervention.

6. PLOS authors have the option to publish the peer review history of their article (what does this mean?). If published, this will include your full peer review and any attached files.

Reviewer #1: No

Reviewer #2: No

Reviewer #3: No

Reviewer #4: **Yes: **Dr. Muthoni E. Karanja

---

## [Author Response · Author response to Decision Letter 0]

8 Apr 2021

PONE-D-20-37236

Impact of Isoniazid Preventive Therapy on Tuberculosis incidence among people living with HIV; a secondary data analysis using Inverse Probability Weighting of patients attending HIV care and treatment clinics in Tanzania.

Journal Requirements:

Response: Thank you very much for the comment. PLOS ONE style has been adopted in the revised manuscript (Lines 1-326).

2. Please include additional information regarding the data extraction tool used in the study and ensure that you have provided sufficient details that others could replicate the analyses. For instance, if you developed a data extraction tool as part of this study and it is not under a copyright more restrictive than CC-BY, please include a copy, in both the original language and English, as Supporting Information, or include a citation if it has been published previously.

Response: Thank you very much for the comment. The study used secondary data collected as part of HIV care and treatment services in the country. Details on how data extraction has been explained in details under methods section (Lines 90-103). The variables to be extracted was guided by planned dummy tables for the manuscript.

"The study was funded through SEARCH Project under Bill and Melinda Gates Foundation. Routine HIV data management is Tanzania is managed collaboratively by the Government of the Republic of Tanzania and the Government of the United States of America through President’s Emergency Plan for AIDS Relief."

Response: Thank you very much for the comment. An updated Competing interests statement has been included in the cover letter as directed.

 Response: Thank you very much for the comment. We have visited the link and read the contents.

Response: Thank you very much for the comment. Detailed explanation regarding data sharing and contact person have been included in the cover letter.

Response: Thanks for the comment. As stated earlier, the permission for data sharing needs to be requested from the Principal Secretary-Health. Upon request, the data set will be made available. The statement regarding data availability has been included in the ethical consideration section (162-164).

Response: Thank you very much for the comment. Ethics statement has been moved into the Methods section (Lines: 152-164).

Reviewers' comments:

Reviewer's Responses to Questions

Comments to the Author

1. Is the manuscript technically sound, and do the data support the conclusions?

Reviewer #1: Partly

Reviewer #2: Yes

Reviewer #3: Yes

Reviewer #4: Yes

2. Has the statistical analysis been performed appropriately and rigorously?

Reviewer #1: No

Reviewer #2: Yes

Reviewer #3: Yes

Reviewer #4: Yes

3. Have the authors made all data underlying the findings in their manuscript fully available?

Reviewer #1: No

Reviewer #2: No

Reviewer #3: Yes

Reviewer #4: No

4. Is the manuscript presented in an intelligible fashion and written in standard English?

Reviewer #1: Yes

Reviewer #2: Yes

Reviewer #3: No

Reviewer #4: Yes

5. Review Comments to the Author

Reviewer #1: In this manuscript, the authors make use of a national dataset to explore the effects of IPT on incident TB among people living with HIV in Tanzania. To address confounding in such an observational dataset, they use an inverse probability of treatment weighting to balance certain confounders between treated and untreated groups. The large size of the dataset and the appropriately chosen method for causal inference are promising, but there are several weaknesses of the analysis as currently presented:

1. The 4 variables adjusted for are not sufficient to balance the IPT and non-IPT groups. IPTW requires balancing all confounders that independently influence both IPT initiation and TB risk. In this dataset, there are a number of other measured but unadjusted confounders that appear to have been associated with IPT initiation (Table 1) and that are also likely to influence TB diagnosis outside of the IPT causal pathway. These include, for example, ART status, BMI and/or nutritional status, and type of health facility. The IPTW analysis is not valid unless all confounders are both measured and included in the weighting, and the authors cannot just assert that “after weighting, the two exposure groups were comparable.”

Response: Thank you very much for the comment. The aim of IPTW is to balance covariates which are more likely to affect treatment selection (in this case is IPT) as well the outcome of interest (in this case is TB incidence). The propensity score model now contains all covariates that are potential confounders; they affect the outcome of interest as well as determine whether a client will be given IPT or not. After weighting the balance of covariates between weighted and original sample was shown quantitatively by using standard differences and variances. (Line 114-134)

The authors also note in the introduction that IPT tended to be introduced sooner at larger facilities, and larger facilities may also be associated with differences in care that influence either risk of developing TB or probability of being diagnosed with TB (e.g. because of different TB diagnostics available), making this another potential confounder.

Response: Thank you very much for the comment. The potential confounding effect due to health facility type was checked. This was checked by including “Type of Health Facility” as one of the covariates. Furthermore, the effect of Type of Health Facility” on TB disease was also checked by carrying out multilevel analysis to check for cluster effect of health facility types. There was no cluster effect detected (Line 147-148).

2. The authors need to do more to describe the motivation and context for their study. While Plos One does not select papers for publication based on “impact”, it is still important to accurately describe the existing literature and what the current analysis adds. The authors start from the assertion that “IPT reduces TB incidence” (abstract), and they cite a number of observational studies demonstrating effectiveness of IPT in reducing TB incidence in routine care settings including Tanzania. They say that the current study will help to inform scale-up, but it is not clear what aspects of scale-up this study informs. Is there clinically important uncertainty in the size of effect based on prior literature? Is there something different about this setting that might lead to a different effect? Are there particular covariates of interest that this analysis is intended to explore?

Response: Thanks for the comment. Sabasaba et al conducted a study to determine effect of IPT in one region in Tanzania. The current study has widened the scope by including three regions. Moreover, the current study unlike most studies before, used IPTW to determine causality between an interventions (IPT) and an outcome or impact of interest (TB incidence). Hence, the study intended to show effectiveness of IPT in routine clinical settings as well as advocate the use of statistical methods to infer causality in observational studies. The motivation and the context of the study has been added in the abstract (Lines 11-12) as well as in the background (Line: 57-69).

Similarly, it seems contradictory in the introduction to state that “a number of studies … in routine public health settings… have also shown benefits of IPT on TB incidence”, but then assert in the next paragraph that “effectiveness [of IPT] in these settings has not been adequately and properly evaluated.” More exploration is needed of what the gaps in current data are and how this study addresses them.

Response: Thank you very much for the comment. The gap was mainly referring to Tanzanian context. Effectiveness of IPT in routine settings has not been extensively conducted. As pointed earlier, Sabasaba et al determined effect of IPT on TB incidence in one region in Tanzania. In other countries like Ethiopia, Brazil and Zimbabwe where IPT implementation has been evaluated, the analysis method was not rigorous as the analysis did not consider the bias introduced by self-selection which is common in observational studies as well as indication bias (confounding by indication) introduced by clinician’s decision as to who to treat and who not to. Hence, the study intended to produce local evidence for Tanzania by involving more regions as well as to show case the use of propensity scores to minimize biases in observational studies for causal inference (Line: 57-69).

3. I note in Figure 2 that a lot of the divergence between the IPT and no-IPT groups occurs within the first two months of follow up. This makes me think that much of the difference is not an effect of the IPT, but rather a difference in the suspicion of existing TB: Clinicians may have been disinclined to start IPT when they suspected that a patient already had active TB, even if an initial diagnostic test was negative or they were still waiting for a diagnostic result. Or, it could be that patients who were going to start IPT got a more thorough evaluation to rule out active TB at baseline, making those with early, undiagnosed TB disease less likely to be included in the IPT group than in the no-IPT group. More information about TB screening and diagnostic practices during the study period might help to clarify this, and it is also a limitation that needs to be explored in the discussion.

Response: Thank you very much for the comment. We agree with you the possibilities of overestimating TB for PLHIV not on IPT during the first days of follow up. This will be included as one of the study limitations (Line: 293-295). 

Minor:

Introduction:

- It’s potentially misleading to cite the % of PLHIV who have TB infection at time of death. Would be more useful to cite the % of PLHIV who have TB disease at time of death.

Response: Thank you very much for the comment. The correction has been done (Line 43-46).

Methods:

- Please state how incidence rates (and their confidence intervals) were calculated.

- How was the multivariable model (Table 3) developed? Why this set of covariates?

Response: Thank you very much for the comment. How incidence rates were calculated and multivariable model was developed has been included in the method section of the manuscript (lines 137-139).

Results:

- Too many significant digits are included, making results both overly precise and difficult to read.

- Person-years do not make sense as reported. Should these be thousands of person years?

- What is the difference between “ambulatory” and “walking” functional status? To me, these seem synonymous.

- Sentences containing lists are difficult to follow due to inconsistent grammar and punctuation.

- Several numerical results seem to be incorrectly typed. For example, incidences by ART status differ between text and table, some confidence intervals are concerning asymmetric, some column percentages in tables do not add to 100%, and some reported p values seem incorrect based on the point estimates and Cis presented.

- TB incidence seems to increase over time from 2012 to 2016. What there a change in diagnostic practice during this period (e.g. introduction of Xpert or change in screening practices), and if so, how is that likely to affect your results? Should calendar year or clinic-level diagnostic availability be adjusted for?

Response: Thank you very much for the comment. Relevant corrections raised have been done. Both calendar year and clinic level were included in the multivariate analysis. Moreover, clinic level associated cluster effect was also checked and had no effect on TB incidence (Lines 145-148).

Discussion:

- I suggest presenting results in the context of follow-up period. IPT reduced TB incidence by xx% over what median duration of follow up? Effects are likely to wane with time, as reinfection increasingly predominates over reactivation as the source of incident TB.

- The main limitation cited is the “inability to include other [confounding] variables”, but they in fact have data on many more potentially-confounding variables that they do have the ability to include if they choose.

- Much more exploration of the study’s limitations is needed.

Response: Thank you very much for the comment. The result section has been reviewed; presentation of results has been changed (Lines 165-222) and study limitations have been reviewed and written as suggested. The limitation was referring to covariates which potentially would be helpful if collected. These were such as health system issues e.g. availability of Isoniazid which is used for IPT (Line 289-295).

Author contributions:

- All authors should review and approve the final manuscript.

Response: Thank you very much for the comment. Contribution of authors has been reviewed (Lines: 316-320).

Reviewer #2: Thank you very much for allowing me to review this interesting manuscript.

The manuscript adds to the existing evidence on the effectiveness of IPT in reducing TB incidence based on analysis of a large number of participants in Tanzania. The papers showed durable protection against TB for 6 years. The manuscript is generally well written but lacks some details. Please consider suggestions below to improve clarity.

1.The definition of the IPT group and the control group is not clear to me. The authors conducted the survival analysis by defining entry time as the data of enrolment. Were participants classified into either IPT group or the control group depending on whether they started IPT at the time of enrolment? I presume that most of PLHIV started at IPT the time enrolment but some might have initiated it later (e.g. after investigation for TB). How did you account for those who started IPT later? For example, if they started IPT one month later, were they then transferred into the IPT group? If they remained in the control group, then the control group is not “those who never received IPT”.

Response: Thank you very much for requesting clarification. The study involved People living with HIV (PLHIV) enrolled from January 2012 to December 2016. The IPT group were those who ever received IPT and the control group constituted those who never received IPT during any time of study duration. For individuals who received IPT the time before receipt of IPT was considered “not on IPT” and the time following initiation of IPT considered “on IPT” (Lines: 105-107) and (Lines: 139-141).

2. What is the definition of TB disease? Only bacteriologically confirmed? Does this include TB diagnosed regardless of the timing after the follow-up? (e.g. TB diagnosed within one week after the enrolment.)

Response: Thank you very much for asking. TB disease diagnosis was either bacteriological or radiological or symptomatic as decided by the clinician. Only PLHIV who were diagnosed with TB before being enrolled into HIV clinics were excluded from follow up. 

3. Figure 2 shows a sharp increase in TB very early in the follow-up. If I understand correctly, participants who screened positive didn’t start immediately upon enrolment and were included in the control group. Among those who screened positive, prevalent TB may have been found after investigation and inflated the number of TB cases in the control group in the initial period.

Response: Thank you very much for the comment. We agree with you TB at the start of follow up may overestimate TB disease among non-IPT group. This has been admitted as a limitation (Lines: 293-295).

4. The authors claim that baseline characteristics were balanced after IPTW. However, I seem to find some imbalances, for example:

ART 29.02% vs 36.09%

Private facility 17% vs 29%

Underweight 15% vs 18%

Is this the reason why the authors conducted multivariable cox-regression after IPTW?

I wonder if the authors conducted balance diagnostics. Presenting results may be helpful. https://www.ncbi.nlm.nih.gov/pmc/articles/PMC4626409/

I would also suggest presenting the distribution of the weights.

Response: Thank you very much for the comment. Balance between the 2 groups (Intervention and control groups) has been quantitatively tested after weighting by using standardized difference and variance between unweighted and weighted samples (Line 125-129).

Minor points

Introduction

Suggest mentioning the coverage of IPT among PLHIV not only the proportion of CTC providing TPT.

Also, suggest to briefly summarize the durability of efficacy reported in high TB burden countries and discuss how the present research adds to the evidence.

Response: Thank you very much for the comment. IPT coverage in Tanzania has also been incorporated (Line 55-56). Efficacy and effectiveness of IPT in other countries have been included (Lines: 57-65).

Method:

Is TST recommended at all?

Response: Thank you very much for requesting clarification. TST is not mandatory for IPT in Tanzania

If I remember correctly, the TB guidelines in Tanzania recommend repeating IPT after 2 years. Is there any chance that some of the participants repeated IPT?

Response: Thank you very much for the comment. Yes, before April 2018 IPT was repeated in every 2 years. However, this analysis did not consider repeated IPT. Individuals were in IPT group if ever received IPT and non-IPT for those who never received IPT (Lines: 105-107).

Data collection

How did you assess nutritional status? What is the definition of each category?

How is ambulatory different from walking?

Do you have information on the previous history of TB?

Response: Thanks for the comment. Nutrition was obtained by calculating Body Mass Index (BMI) from weights and heights of clients.). The definitions were according to World Health Organization as follows: Underweight (<18.5kg/m2) Normal (18.5kg/m2), overweight (25 kg/m2) and obese (>30 kg/m2). 

The study participants were newly enrolled PLHIV from January 2012 to December 2016. Those with TB before enrolment in HIV clinics were excluded from the cohort. Prior TB apart from TB with which a client reported to HIV clinic was not known.

The right categories are bedridden, ambulatory and working. The right correction has been done.

Analysis

Please justly the use of p-values less than 0.05 as a criterion to select covariates included in the final model. This may miss important variables associated with exposure and outcome. Have you considered a higher threshold? https://pubmed.ncbi.nlm.nih.gov/8256780/

Response:

How did you handle missing data?

Response: Thanks for the comment. The threshold for including a covariate into a multivariate model was lifted to P-value of a maximum of 0.2 (Lines 144-145) to include more covariates as suggested in the article given for reference. The distribution of missing was checked. Missing data had no effect on the outcome of interest; they were at random. If they were systematic we would apply manipulations such as imputation. Hence, the data analysis was done using the whole dataset.

How were the weights given? Was it simple inverse or did you use stabilized weights?

Response: Thanks for the comment. The weights were given using simple inverse (Line 129-134).

Result

Paragraph 2

“from private health facilities (84.36%).” This is not consistent with Table 1. 84.36% is from public facilities.

Response: Thank you very much for the comment. This has been rectified.

Paragraph 3

Suggest reporting the median duration of follow-up in the two groups. The person-years reported don’t seem to correct. Only 134.56 person-year? Is this for all participants or 2309 TB patients? Either way, the number is too small, corresponding to a very short period of follow-up. Person-years in Table 2 are also too small.

Did you find any difference in the number of people who were lost to follow-up or died between the two groups? How could that impact the analysis?

Response: Thank you very much for the response. The follow up duration in each of the group has been shown in the result section. Lost to follow up was not considered in this analysis.

Discussion

Paragraphs 2 and 3 seem repetitive. The discussion could also mention the durability of protection.

Response: Thank you very much for the comment. This has been corrected. In the discussion section.

Table 1.

Proportions of the weighted sample enrolled by year in the IPT group do not sum up to 100%.

Response: Thank you very much for the comment. This has been corrected

Reviewer #3: There is no statement on the ethical consideration that explains how to keep the confidentiality of the participants. Try to include a statement. It also needs some grammatical error corrections. both on the introduction and discussion.

Response: Thank you very much for the comments. Ethical consideration kept participants’ confidentiality by using only unique identification instead of their names. Ethical clearance has been reviewed to reflect the comment (Line 152-164).

Reviewer #4: ‘Impact of Isoniazid Preventive Therapy on Tuberculosis incidence among people living with HIV; a secondary data analysis using Inverse Probability Weighting of patients attending HIV care and treatment clinics in Tanzania.

This manuscript is intended to describe the use of inverse probability weighting (IPW) to determine the effectiveness of IPT in preventing active TB among PLHIV attending health facilities in Tanzania. The manuscript is good addition to scientific literature as it demonstrates the benefit of using IPW and its results. Previous studies on impact of TB prevention and TB Incidence in Tanzania has been done but the uniqueness of this paper is that it has included children, more regions in addition to Dar es Salaam and use of IPW to determine TB incidence among those on TB Prevention. The author needs to define and clearly state the objective of this study.

Response: Thank you very much for the comment. The objective of the manuscript has been revised Back (Lines 14-15, 71-74).

The abstract section is well written and reflecting on the manuscript. However, the author has indicated 171,672 as the study participants but this is not reflected in the results section which indicates 166,709 can this be clarified. The background section seems to be missing the objective and the statement on routine settings should be clear they should consider routine care setting. The conclusion should be aligned to the study objectives.

Response: Thank you very much for the comment. Abstract, background, result, discussion and conclusion sections are aligned.

In the background section, in the 2nd paragraph the author has indicated Tanzania adopting WHO 3 I's in 2011 , can this be expounded was it in the form of guidelines or policy directive from the ministry ? and further expound on the scale up was it phased from higher level to lower level .The author has indicated > 50% scale up in the CTCs can this indicated in patient numbers as well .The policy on IPT uptake is it once in a lifetime or repeated after a certain duration of time in Tanzania .The fourth paragraph should read limited information on effectiveness of IPT on TB incidence as there has a study published on this (Sabasaba et al. BMC Infectious Diseases)

Response: Thank you very much for the comment. We have worked on the above (Lines 37-74). Tanzania started implementing 3Is in 2011. IPT uptake was repeated in every 2 years before April 2018. After April 2018, IPT was given only once.

The manuscript would have benefitted from a clearer, detailed and a logical methodology section for the readers to understand. The authors should consider to elaborate on the study design, clearly describe the study setting or a brief on the 3 regions. 

.An explanation on the TB/HIV service delivery in particular IPT services in the health facilities in this 3 regions including TB screening process , diagnosis , treatment and follow up has also been included in the manuscript .The authors needs to be clear on the study population .In the data analysis section the author needs to clearly indicate how the 4 variables ( Age,sex, region WHO staging ) was achieved at, justify why IPW .A clear explanation on the propensity scores used and how this was achieved .There should be a step wise approach with clear formulae and outcomes .

Response: Thank you very much for the comment. A brief explanation on the TB/HIV service delivery in particular IPT services in the health facilities in this 3 regions, the TB screening process, diagnosis, treatment and follow up has been included in the manuscript (Lines: 78-89). The study population has been clearly stated (Lines: 105-107). How the propensity score model was constructed has been explained (Lines: 114-134). Justification for using IPTW has been given (Lines 69-74). Steps involved in the IPTW have been included in the manuscript (Lines:114-134). Formulae and outcomes used are shown below:

1. Notations for logistic regression for selection of covariates:

Logit(IPT)=a+b(Sex)+b(Age)+b(Functionalstatus)+b(ART)+b(BMI)+b(Nutrition)+b(HealthFacilitytype)+b(Region)+b(Health Facility ownership)+e

Logit(TB disease)= a+b(Sex)+b(Age)+b(Functional status)+b(ART)+b(BMI)+b(Nutrition)+b(Health Facility type)+b(Region)+b(Health Facility ownership)+e

Where a=Outcome at baseline, b=co-efficient of covariates, e=error term

2. Notations for propensity scores:

ATE=Expected ((IPT)-(Non-IPT))

Propensity Score (Probability of receiving IPT given a certain covariate), PS=Prob(IPT/Covariate)

3. Notations for model balance: 

D=Prop(IPT)-Prop(Non-IPT)/√(PropIPT(1-PropIPT)+PropNon-IPT(1-Non-IPT))/2

 4. Notations for propensity score weights:

PS weights=1/PS + 1(1-PS)

Where, 1/PS=Weight for IPT group, 1/(1-PS) =Weight for Non-IPT group

In the result section, the authors need to have a clearly structured sub section of the results. There needs to be clarification on the actual number of study participants analyzed (171,743 or 171,672 or 166,709?) and let it be uniform throughout the document. Table 1 is not uniform and a brief explanation on the age stratification from 0-9 then 10 – 19 then 20 – 24 then 25- 29. There needs to a clear write up of the highlight summary findings of the 3 tables in the result section and revision of the 3 tables as they appear overcrowded.

Response: Thank you very much for the comment. The result section has been revised to accommodate the suggested comments

In the discussion section, in the first paragraph the aspect of using IPTW to determine effectiveness of public health intervention has been previously used refer to the NIH study (Sheri A. Lippman et al .NIH ) it’s more of use of TB prevention and in this country setting that is limited .The author seems to have duplicate information on ‘IPT lowering TB incidence by 70% ‘ in the 2nd and 4th paragraph can this be revised .From the commencement of TB prevention in the HIV facilities from 2011 has the national TB program seen some changes like reduction in the TB/HIV co-infection rate ? . Can the author expound on the association of higher TB incidence in the variables: Male; not on ART; aging population; Nutrition status; higher numbers in 2016 and in the Dar es salaam region and the public health facilities of TB incidence. Basically the authors need to further expound on the findings of the current study in line with the results they got.

Thank you very much for the comment. Relevant corrections have been done in the discussion section.

Finally, perhaps the authors can strengthen the conclusion in line with the objective of the study perhaps inclusion on use of IPW in evaluating effectiveness of a public health intervention.

Thank you very much for the comment. The conclusion section has been revised (Lines: 296-299).

6. PLOS authors have the option to publish the peer review history of their article (what does this mean?). If published, this will include your full peer review and any attached files.

Do you want your identity to be public for this peer review? For information about this choice, including consent withdrawal, please see our Privacy Policy.

Reviewer #1: No

Reviewer #2: No

Reviewer #3: No

Reviewer #4: Yes: Dr. Muthoni E. Karanja

Thank you very much for the comment. Figures have been uploaded in PACE.

Comments on the Plos One paper:

The paper attempted to study a very important question, well done for the initiative. Please find my following comments regarding the article below.

Response: Thank you very for the comment.

Abstract

Background: there is no statement that explains the magnitude of the problem (statement of the problem).

Response: Thank you very much for the comment. The magnitude of TB among people living with HIV has been included (Line 43-46)

Methods:

study setting and design not included on the methods section of the abstract, when are you going to say the IPT has no effect, reduces the effect or increases the effect, the decision criteria is not set on this section. Or the cutoff point is not placed here. Try to include it. In the methods of the abstract, some of the sections of the method of the abstract does not make sense it needs revision. 

In the result of the abstract, one change that should be made revolves around the Results sections on the "incidence" is that incidence or prevalence??

Response: Thank you very much for the comment: The conclusion of effectiveness of IPT on reducing TB disease among PLHIV has been drawn in comparison with studies from other settings. There is no cutoff point. Relevant changes have been made in the abstract section to include study setting and design.

Data collection: the data collection technique is not included on the data collection section try to include it there. 

Response: Thank you very much for the comment. Data collection technique is included (Lines 90-103).

Methods section:

Please clearly elaborate the following procedures: inclusion and exclusion criteria, sampling, methods for data capture and quality control. Any attempt to reduce internal and external biases, which confounders were considered? There is missing of eligibility criteria, Data collection, Handling and Tracking of Missing data, and Quality control.

Response: Thank you very much for the comment. The study included PLHIV enrolled in HIV clinics from January 2012 to December 2016 in 3 regions in Tanzania. The study excluded from analysis, PLHIV who had existing TB disease (on treatment) before HIV clinic enrolment (Lines 100-103). 

The regions were conveniently sampled as they had high HIV and TB incidence. As data used for analysis were those of routine practices, there are mechanisms to make sure that routine data are of quality at routine program level. These include training of staff and ongoing capacity building through mentorship, supportive supervision and data quality assessment. Different measures have been taken to minimize biases. These include propensity score weighting and multivariate analysis. Missing data were examined and were not found to have significant effect and thus analysis was done for the whole sample.

Ethical consideration

There is no statement on the ethical consideration that explains how to keep the confidentiality of the participants. Try to include a statement.

Response: Thank you very much for the comment. A statement on study participant confidentiality has been added in the ethical consideration section (Line 153-164).

Discussion

You should include a paragraph in the discussion about the significance of your results and how this will inform preventative strategies in the future.

Response: Thank you very much for the comment. A sentence on the implication of study findings on the practice/policy on TBHIV prevention has been included (Line 278-283).

---

## [Decision Letter · Decision Letter 1]

26 Apr 2021

PONE-D-20-37236R1

Impact of Isoniazid Preventive Therapy on Tuberculosis incidence among people living with HIV; a secondary data analysis using Inverse Probability Weighting of individuals attending HIV care and treatment clinics in Tanzania.

PLOS ONE

Dear Dr. Maokola,

Thank you for submitting your manuscript to PLOS ONE. After careful consideration, we feel that it has merit but does not fully meet PLOS ONE’s publication criteria as it currently stands. Therefore, we invite you to submit a revised version of the manuscript that addresses the points raised during the review process.

We look forward to receiving your revised manuscript.

Kind regards,

Katalin Andrea Wilkinson, PhD

Academic Editor

PLOS ONE

Journal Requirements:

Reviewers' comments:

Reviewer's Responses to Questions

**Comments to the Author**

1. If the authors have adequately addressed your comments raised in a previous round of review and you feel that this manuscript is now acceptable for publication, you may indicate that here to bypass the “Comments to the Author” section, enter your conflict of interest statement in the “Confidential to Editor” section, and submit your "Accept" recommendation.

Reviewer #2: (No Response)

Reviewer #4: (No Response)

2. Is the manuscript technically sound, and do the data support the conclusions?

Reviewer #2: Yes

Reviewer #4: Yes

3. Has the statistical analysis been performed appropriately and rigorously? 

Reviewer #2: Yes

Reviewer #4: Yes

4. Have the authors made all data underlying the findings in their manuscript fully available?

Reviewer #2: No

Reviewer #4: Yes

5. Is the manuscript presented in an intelligible fashion and written in standard English?

Reviewer #2: Yes

Reviewer #4: Yes

6. Review Comments to the Author

Reviewer #2: Thank you very much for revising the manuscript.

I only have a few minor comments below.

Line 100: “TB infection” usually refers to LTBI in the context of TB prevention and thus confusing.

Line 106: What do you mean by “at least one cycle”. Do you mean completion of 6-month? Isn’t all who started IPT included?

Lines 121-123 and Lines 130-132.

Reviewer #4: I appreciate the opportunity to review this manuscript which describes the use of Inverse Probability for Treatment Weighting (IPTW) to determine the relationship between IPT and TB incidence among PLHIV attending care. This will be a great addition especially in minimizing bias in non-experimental studies. The authors have addressed a number of corrections and suggestions which is very commendable. A few considerations to be looked into by the authors are as summarized below:

Abstract

The authors need to consider revising the background section its more of a statement rather a recommendation. The objectives are not smart as well. Line 33-34 is not necessary can the authors consider revision of that sentence.

Discussion

The authors should consider revising line 239 -244 a bit of repetition. They have explained some of the results as written in line 257 – 270 however, they need to further expound on the TB incidence rate why it is lower in private health facilities as depicted in the results section as well as the reason it is lowest in 2012 and increasing throughout the study.

Minor

The authors need to work on the grammatical errors and omissions throughout the whole document.

7. PLOS authors have the option to publish the peer review history of their article (what does this mean?). If published, this will include your full peer review and any attached files.

Reviewer #2: No

Reviewer #4: No

---

## [Author Response · Author response to Decision Letter 1]

14 Jun 2021

Response to reviewers addressing all the comments has been attached in this submission as suggested and labelled "Response to reviewers"

---

## [Editor Report · Decision Letter 2]

21 Jun 2021

Impact of Isoniazid Preventive Therapy on Tuberculosis incidence among people living with HIV; a secondary data analysis using Inverse Probability Weighting of individuals attending HIV care and treatment clinics in Tanzania.

PONE-D-20-37236R2

Dear Dr. Maokola,

We’re pleased to inform you that your manuscript has been judged scientifically suitable for publication and will be formally accepted for publication once it meets all outstanding technical requirements.

Kind regards,

Katalin Andrea Wilkinson, PhD

Academic Editor

PLOS ONE
---

## [Editor Report · Acceptance letter]

28 Jun 2021

PONE-D-20-37236R2 

Impact of Isoniazid Preventive Therapy on Tuberculosis incidence among people living with HIV; a secondary data analysis using Inverse Probability Weighting of individuals attending HIV care and treatment clinics in Tanzania. 

Dear Dr. Maokola:

I'm pleased to inform you that your manuscript has been deemed suitable for publication in PLOS ONE. Congratulations! Your manuscript is now with our production department. 

Kind regards, 

on behalf of

Associate Professor Katalin Andrea Wilkinson 

Academic Editor

PLOS ONE